# PLGA Particles in Immunotherapy

**DOI:** 10.3390/pharmaceutics15020615

**Published:** 2023-02-11

**Authors:** Dennis Horvath, Michael Basler

**Affiliations:** 1Division of Immunology, Department of Biology, University of Konstanz, D-78457 Konstanz, Germany; 2Centre for the Advanced Study of Collective Behaviour, University of Konstanz, D-78457 Konstanz, Germany; 3Biotechnology Institute Thurgau (BITg) at the University of Konstanz, CH-8280 Kreuzlingen, Switzerland

**Keywords:** antigen-delivery systems, nanoparticles, microparticles, PLGA, controlled release, drug repurposing, immunotherapy, vaccination, biodegradable

## Abstract

Poly(lactic-co-glycolic acid) (PLGA) particles are a widely used and extensively studied drug delivery system. The favorable properties of PLGA such as good bioavailability, controlled release, and an excellent safety profile due to the biodegradable polymer backbone qualified PLGA particles for approval by the authorities for the application as a drug delivery platform in humas. In recent years, immunotherapy has been established as a potent treatment option for a variety of diseases. However, immunomodulating drugs rely on targeted delivery to specific immune cell subsets and are often rapidly eliminated from the system. Loading of PLGA particles with drugs for immunotherapy can protect the therapeutic compounds from premature degradation, direct the drug delivery to specific tissues or cells, and ensure sustained and controlled drug release. These properties present PLGA particles as an ideal platform for immunotherapy. Here, we review recent advances of particulate PLGA delivery systems in the application for immunotherapy in the fields of allergy, autoimmunity, infectious diseases, and cancer.

## 1. Introduction

Drug delivery of active pharmaceutical ingredients (APIs) into the body is crucial for a successful therapy. Ideally, targeted delivery of the active compounds to the desired site of action is intended to reduce adverse side effects and at the same time increase drug efficacy by higher local concentrations of APIs. To fulfill this role in targeted drug delivery, a multitude of synthetic polymers have been developed as drug carriers over the recent years. Alongside lipid nanoparticles, poly (lactic-*co*-glycolic acid) (PLGA) particles in the nano (NP) and micron (MP) range are the most commonly used and extensively studied drug carrier polymers for biomedical applications [1]. PLGA formulations exhibit a range of favorable properties for usage in biologic systems such as good bioavailability and biocompatibility, a biodegradable polymer backbone, and a prime safety profile. Hence, PLGA is approved by the U.S. Food and Drug Administration (FDA) and the European Medicines Agency (EMA) for parenteral or mucosal applications in humans [2]. Drugs can be adsorbed to and encapsulated into the particle core or the polymer matrix as well as conjugated to the building blocks or the particle surface. Thereby, APIs can be shielded from degradation and premature clearance from the circulation and in addition conceal their possibly inconvenient smell and appearance [3]. At present, over 20 PLGA particle-based formulations are approved for commercialization, most of which present delivery of chemotherapeutics for the treatment of malignant tumors. Moreover, a large number of different biomedical applications for PLGA particles are being investigated in preclinical and clinical studies ranging from treatment of cardiovascular and inflammatory diseases to wound healing and tissue regeneration [4,5,6].

In recent years, immunotherapy has emerged as an impactful treatment option in a variety of diseases. Most notably, the encouraging success of immune checkpoint inhibition therapy in melanoma patients shifted the focus in medical research towards the development of novel immunotherapies [7]. The term “immunotherapy” describes the treatment of diseases via modulation of the immune system. It has been shown to be effective in a progressively increasing number of different diseases. In comparison to standard therapies, immunotherapy features certain benefits such as higher treatment efficacies with likewise fewer adverse side effects. Thereby, controlling immune responses via introduction of activating or silencing factors in a targeted manner is the key goal of effective immunotherapy [8,9].

PLGA particles exhibit ideal properties as platforms for targeted delivery of immunotherapeutic drugs. Via controlling their starting material, preparation methods, and conditions, various particle shapes can be designed including spheres, capsules, tubes, and gels, adapted to the loaded drugs and targeted organs or cell types [10,11]. Further, PLGA particles can easily be modified by surface coating or chemical conjugation of targeting molecules and active substances. Thus, PLGA particles can be tailored for directed delivery and controlled release of immunomodulatory compounds to very specific parts of the immune system [12]. Capitalizing on these properties of PLGA particulate delivery systems, researchers have made progress in the development and optimization of particle-based immunotherapy. In this article, we review recent advances in the application of PLGA particles for immunotherapy in the treatment of allergic diseases, autoimmune diseases, infectious diseases, and cancer.

## 2. Properties of PLGA Particles

An array of substances including biologically active compounds such as proteins and peptides, nucleic acids, as well as immunomodulatory molecules can be incorporated into, adhered to, or conjugated as cargo to PLGA particles [13,14,15,16,17,18,19,20,21,22]. Thereby, the cargo substances are either entrapped into the polymer matrix of, e.g., spheric particles, incorporated into the core of capsules, or adhered to the particle surface. Further, surface modifications of PLGA particles can be utilized to conjugate cargo molecules (Figure 1). Thus, depending on the delivery method, cargo molecules can be shielded from or explicitly exposed to the surrounding tissue. This allows for specific and efficient targeting of desired immune cells and exhibits a range of advantages compared to soluble drug formulations. First and foremost, the controlled release of cargo from PLGA particles can be adjusted from several hours to months of sustained supply with loaded substances. The release of compounds from PLGA particles is commonly characterized by a typical tri-phasic release profile, though bi- or mono-phasic release patterns are reported as well [23]. The first phase represents an initial burst, attributed to weakly encapsulated cargo or adherence of the cargo to the particle surface. This is followed by the second phase—a slow, continuous release over time, mediated by the progressing hydrolysis of the polymer matrix. The third phase exhibits a final burst release of the remaining substances, due to erosion of the particle [23,24,25].

One approach influencing the release kinetics is the adjustment of polymer degradation. In aqueous milieu, the PLGA polymer backbone undergoes slow non-enzymatic hydrolysis and is degraded over time into its monomers lactic and glycolic acid (Figure 2). The end-product monomers are completely metabolized in the citric acid cycle and removed from the body [26]. Degradation kinetics can be influenced by adjusting the lactic:glycolic acid ratio of the polymer. In general, a higher content of lactic acid yields a slower degradation and, thus, a slower release of the cargo over time due to increased hydrophobicity [27,28]. Further, the ratio of lactic:glycolic acid impacts the crystallinity of the co-polymer. While lactic acid is crystalline, glycolic acid exhibits more amorphous properties. Higher contents of glycolic acid will shift the ratio of crystalline:amorphous towards more amorphous regions within the formulated particles, which results in faster particle hydrolysis. The fastest degradation rate is achieved with a 50:50 ratio of lactic to glycolic acid [29,30]. The molecular weight of the polymers used for the manufactured particles has an impact on the particle degradation as well. With higher molecular weights, the polymeric chain size increases and decelerates the degradation rate as a consequence [31]. Particle size plays another important role in defining the release profile of PLGA particles. Smaller particles, especially in the nanoparticle range are associated with a higher initial burst and a general faster release of the carried components compared with larger particles in the micron range [32,33]. Further, drug loading itself has a significant influence on the release profile. PLGA particles with higher initial amounts of cargo loaded exhibit an increased release rate [34]. The characteristics of the loaded substances have also been reported to influence the release profile. More hydrophilic compounds show a faster release profile as their release from waterfilled pores is facilitated compared to more hydrophobic ones [35,36,37]. Other properties of PLGA particles have also to be considered when adjusting the release patterns, i.e., the glass transition temperature (T_g_), pH of the release medium, and the surface charge of the particles [38,39]. The particle properties can also be influenced by the applied preparation method. Several methods for particle formulation have been described, with the most commonly used being the emulsification–solvent evaporation technique. Depending on the drug characteristics, two variants of this technique are available: hydrophobic compounds can be incorporated via oil in water single solvent evaporation method, while hydrophilic compounds must be encapsulated in a double emulsion. Other preparation methods used for PLGA particle formulation encompass spray-drying, phase separation, microfluidics, PRINT technology or nanoprecipitation. Particle preparation methods and parameters are reviewed elsewhere in detail [1,11,38,40,41,42,43].

For the applications in immunotherapy, PLGA particles must be designed for interactions with cells of the immune system. The immune cells with the highest priority to be targeted are professional antigen presenting cells (APC), in particular macrophages and dendritic cells (DC). Here, DCs are of outstanding importance as they represent the link between the innate and adaptive immunity with their unique ability to prime naïve T cells to induce a full-blown antigen-specific immune response [44]. Researchers harness different strategies for drug delivery by PLGA particles to immune cells. DCs and macrophages show highly phagocytic properties and are able to engulf PLGA particles via macropinocytosis, clathrin-dependent receptor-mediated endocytosis and phagocytosis [45]. Here, the particle size is the deciding crucial factor for efficient uptake of PLGA particles. In general, PLGA nanoparticles as well as microparticles up to a size of 5 μm in diameter (compare Figure 3) were shown to be efficiently taken up by DCs [46,47,48] as this range also resembles the magnitude of most pathogens. Upon endocytosis, particle cargo is released into the endosome. Acidification of the endosome via fusion with the lysosome can accelerate particle degradation and compound release. In case of antigenic cargo, processing by lysosomal proteases will supply antigenic peptides for antigen-presentation to CD4^+^ T cells on major histocompatibility complex (MHC) class II molecules [49]. In addition, PLGA particles have been reported to escape from the endosomal compartment into the cytosol [47,50]. Therefore, antigen-processing by the proteasome and subsequent cross-presentation to CD8^+^ T cells on MHC class I is ensured [50,51]. Furthermore, immunomodulatory compounds, delivered by the PLGA particles can reach their receptors, e.g., toll-like receptor (TLR) ligands, in the endosome as well as in the cytosol [52,53]. In fact, PLGA particle-mediated antigen delivery was shown to significantly increase the presentation of both MHC class I and II peptides [54,55]. Additionally, it was demonstrated that especially PLGA particles in the micron range exhibit immunostimulatory activity [56]. The intrinsic adjuvant activity of PLGA MP was attributed to NACHT, LRR and PYD domains-containing protein 3 (NLRP3) inflammasome activation in DCs upon particle uptake [57]. This characteristic can circumvent the use of additional immunostimulatory molecules and thereby reduce the drug burden in PLGA particle-based immunotherapy. On the other hand, this carries the risk of unwanted inflammation and an inappropriate immune response.

Besides directly affecting the uptake by DCs and macrophages, particle size exhibits another important factor namely particle distribution upon administration into the body. Nanoparticles are able to drain via the lymphatic vessels to the draining lymph nodes (dLN), where they can be taken up by lymph node resident DCs. Larger particles do reside at the site of administration exhibiting a local depot for the loaded cargo. The drugs can then be released in the peripheral tissue and passively spread to dLN and/or the particles can be taken up by DCs and subsequently transported to the secondary lymphoid organs such as the dLN. Noteworthy, nano-scale PLGA particles can reach a systemic distribution and can be tailored to even pass physiological barriers such as the blood–brain barrier (BBB) to deliver the loaded substances, e.g., to the central nervous system (CNS) which, depending on the application, can have advantageous effects [58]. However, the hard-to-control non-specific distribution of polymeric nanoparticles is still a problem to be solved by researchers to avoid nanotoxicology in patients [59,60]. Although particle size is an important factor in the design of PLGA particles for efficient uptake by immune cells, other parameters such as zeta potential and surface charge, hydrophilicity, and particle morphology [56,61,62] need to be considered as well. As a matter of fact, adjusting the parameters to optimize particle uptake crucially dictates the efficacy of the fabricated particle in modulating an immune response. In a study by Look et al., administration of PLGA nanoparticles encapsulating the immunosuppressive mycophenolic acid failed to achieve amelioration of clinical scores in a mouse model of lupus erythematosus compared to vesicular nanogels. The inferior therapeutic impact on the disease was attributed to lower cellular uptake by DCs compared to the nanogel counterpart [63]. To circumvent limitations in PLGA particle efficacy for immunotherapy, researchers have developed methods to improve targeted particle delivery and uptake by cells associated with the immune system. Most manufactured PLGA particles exhibit negative surface charges. However, to interact with the cell membranes of immune cells to facilitate phagocytosis, positively charged particles are beneficial [64,65,66]. Thus, surface modifications of PLGA particles using positively charged polymers such as polyethylenimine (PEI) [67,68] or chitosan (CS) [69,70,71] are commonly performed in order to shift the zeta potential towards positive. The hydrophobic nature of PLGA particles presents a challenge for the use in immunotherapy. Hydrophobic particles are rapidly cleared from the organism by the reticuloendothelial system (RES). In order to increase the half-life of particles in the circulation via concealing the particles from the RES and avoid unwanted adhesions of the hydrophobic polymers, surface modifications are introduced that generate a hydrophilic environment around the particles [72]. The most common agents used for this kind of surface modification are polyethylene glycol (PEG) [73,74], polyethylene oxide (PEO) [75], and poloxamers [55,76]. Especially PEGylation of PLGA particles has been extensively studied and is consistently utilized for particle modifications. It has been shown that coating with PEG benefits PLGA particle drug delivery. PEGylation is associated with a gain in immunologically relevant functions such as enhanced interaction with components of the blood and facilitated muco-adhesion in the gut associated lymphoid tissue (GALT) [77].

Besides general optimization of particle characteristics for immunotherapy, research is conducted on surface and polymer modifications to achieve targeted drug delivery by PLGA particles to specific cell types. As an example, coating of PLGA particles with T cell specific antibodies or antibody F(ab’) fragments achieved targeted delivery of encapsulated immunomodulatory substances to CD4^+^ and CD8^+^ T cells ex vivo and in vivo in a murine tumor model [78,79,80]. Pei and colleagues coated PLGA nanoparticles with exosome membranes to target drug delivery to target M2 type macrophages in the airways as a treatment for asthma [81]. These studies show the potential of targeting PLGA particles to specific immune cell subsets. Especially in immunotherapy, it is important to address a certain part of the immune system to induce a therapeutic response or eliminate a dysregulation in immunity, while at the same time ensure immune homeostasis and avoid potential devastating side effects. Another way of dictating site-directed immune responses and therapeutic immunomodulation is choosing the appropriate route of administration. PLGA is approved by the FDA and the EMA for parenteral administration. Here, the site of injection controls the phagocytic cell type that internalizes the particles. While resident DCs are the main phagocytes for PLGA particle uptake upon intramuscular, intra dermal, or subcutaneous injections, intra peritoneal injections result predominantly in phagocytosis by macrophages [82]. At the same time, the induction of T helper (Th) 1 versus Th2-mediated immune responses as well as the efficacy of the induced response can be tailored by deciding to the route of injection [83,84]. Alternatives to parenteral applications of PLGA particles represent the mucosal administration via the oral, or intra nasal route. Though orally administered PLGA particles are subjected to the highly acidic environment of the stomach and subsequently to highly accelerated polymer degradation, it has been reported that PEGylated particles are suited to pass the gastrointestinal tract for site directed delivery of cargo [85,86,87]. Intranasal applications are of interest for drug delivery directed to the CNS and the generation of site-directed immune modulation in the bronchoalveolar tract which represents the main entry point for many pathogens causing infectious diseases as well as allergens [88,89].

In summary, the design and the application of PLGA particles highly influence the particle distribution in biologic systems, the uptake by immune cells, and subsequently the modulation of the immune response. The often times intertwined effects of PLGA particle properties on these parameters are summarized in Table 1.

## 3. Most Successful Application of PLGA Particles in Immunotherapy

### 3.1. Allergic Diseases

The development of allergic disease is strongly associated with a western lifestyle with incidences also on the rise over the Asian continent. The pathology of allergic diseases can have severe impacts on patients’ health and quality of life [90]. The most commonly occurring type I allergy is characterized by a predominantly Th2 mediated inappropriate immune response against effectively harmless antigens. Upon antigen encounter, surface-bound IgE initiates rapid degranulation of mast cells and basophils and subsequent infiltration and activation of other immune cells such as eosinophils and Th2 cells. In severe cases, this can lead to life-threatening conditions by triggering anaphylactic shocks [91]. Hence, the development of novel technologies for the treatment and prevention of allergies is of high priority. PLGA particle mediated immunotherapy presents an ideal tool for suppressing allergic disease and has been established as a potential treatment option in various allergies [92].

Allergic asthma is one of the most prevalent allergic diseases. It is mainly treated with anti-inflammatory medication to counteract the immune imbalance [93]. PLGA particles have been investigated as carrier molecules for delivery of anti-inflammatory agents as potential treatment options for allergic asthma.

Encapsulation of antisense oligonucleotides into PLGA nanoparticles for gene targeting of DNA methyltransferase 3 (Dnmt3), a key factor for differentiation of M0 to M2 macrophages, yielded effective silencing of M2 macrophages in a mouse model for allergic asthma. To efficiently target M2 macrophages, the group modified the Dnmt3aos silencer carrying particles with exosome membranes which have been generated in vitro of bone marrow derived macrophages. These engineered particles achieved exceptional delivery to macrophages in vitro and in vivo, resulting in Dnmt3aos gene silencing with reduced protein expression and a decrease in M2 macrophage population and subsequent mitigation of inflammation in the lung [81].

Anti-inflammatory flavonoid chrysin was investigated for formulation into PLGA nanoparticles for treatment of allergic asthma. Encapsulation of chrysin was shown to circumvent the shortcomings of the flavonoid such as poor bioavailability and solubility in aqueous milieus. Treatment of mice subjected to ovalbumin (OVA)-induced allergic asthma with chrysin-nanoparticles ameliorated the disease symptoms. The effect was attributed to repression of the NLRP3 inflammasome activation and, thus, lower induction of allergy specific immune parameters such as serum IgE titers and Th2 cytokine levels [94].

Despite perpetual progress in the development of novel treatment options for asthma including PLGA-based immunotherapy, hypo sensitization via allergen specific immunotherapy (SIT) remains the gold standard in the treatment for allergic disease. SIT aims at shifting the predominantly Th2 mediated immune response against allergens towards a Th1 type response, thereby reducing the IgE response and substituting it with IgG (preferably IgG2a or IgG4) blocking antibodies. Further, induction of peripheral tolerance to the allergen is desired, mediated by interleukin (IL)-10 secretion of regulatory T cells (Tregs) [95]. In SIT, desensitization is achieved with regular exposure to increasing doses of the target allergen. Though effective therapy is obtained eventually, SIT is accompanied by high costs due to a multitude of up to 80 injections over several years and unpleasant allergic side effects [96]. SIT exploiting the properties of PLGA particles can circumvent the aforementioned disadvantages of the therapy. With the sustained release and tunable properties of PLGA, multiple injections with antigens become obsolete. Noteworthy, immune responses elicited by PLGA particle-mediated vaccinations tend towards a Th1 type response compared to default adjuvants such as aluminum salts which are known to induce Th2 biased responses. The shift to a Th1 immune response can further be constrained by co-encapsulation of immunostimulatory adjuvants together with antigens into PLGA particles [97].

Indeed, PLGA particle-mediated SIT accomplished inhibition of allergic asthma in mice. In a study conducted by Lou et al., a PLGA nanoparticle vaccine was formulated with OVA and the E3 ligase A20. Upon vaccination, the Th2 type inflammatory response produced by the allergic asthma model could be prohibited, and an increase in Treg populations and IL-10 secretion was assessed [98]. The generation of a PLGA-PEP-PLGA hydrogel as antigen depot for SIT showed conformable results. Allergen-sensitized mice that were treated with PLGA-PEP-PLGA-based SIT exhibited comparable ameliorative effects in asthma symptoms as mice treated with standard SIT. Concerning both immune parameters, hydrogel SIT and standard SIT achieved a reduction in Th2 cell abundance and IgE titers. However, only the PLGA-PEP-PLGA-mediated therapy resulted in increased antigen specific IgG1 levels [99]. In general, PLGA particle-mediated SIT was shown to be effective in several allergic diseases of the airways including allergic conjunctivitis and allergy induced by house dust mites (HDM). Encapsulation of allergic proteins and co-encapsulation of immunostimulants into PLGA particles is an elegant method to shift allergen-specific immune globulin production from IgE isotypes towards IgG2a and stimulate a more favorable Th1 type immune response [100,101].

Mucosal exposure to allergens other than via the airways occurs most notably in the gastrointestinal tract. Food allergies depict a popular health problem with increasing incidences over the last decades, and the need for preventive options emerges [102]. Recent studies on PLGA particle-mediated prophylaxis for the development of cow’s milk allergy (CMA) have indicated a practice of PLGA in the field as carrier for food allergens. In mouse models of whey-protein induced CMA, oral application of PLGA nanoparticles loaded with β-lactoglobulin derived peptides protected C3H/HeOuJ mice from anaphylaxis and attenuated allergic skin response. Secretion of the systemic pro inflammatory cytokines IL-6 and tumor necrosis factor (TNF)-α upon whey was prevented when mice were desensitized with peptide loaded PLGA nanoparticles and in turn, immune homeostasis was preserved [103,104]. Noteworthy, β-lactoglobulin-loaded microparticles have shown efficacy in oral tolerance induction in mice with a single-dose administration and effectively decreased β-lactoglobulin specific serum IgE even with a 10^4^ magnitude lower dose than that of what was required using soluble allergens [105,106]. These findings underline the benefits of PLGA particle-mediated immunotherapy targeting allergy, i.e., safer delivery of antigens due to lower dosing and the circumvention of multiple applications.

Allergies can develop from a vast variety of origins. One common source of allergens is insect-derived venoms such as the honeybee venom. Insect stings can ignore the natural barrier of the skin and can apply the allergens directly into the system. Thus, a single exposure can lead to rapid induction of a life-threatening anaphylaxis in allergic individuals [107]. For the use of PLGA particles as SIT against honeybee venom allergy, encapsulation of bee venom or venom components have been investigated. Successful encapsulation and release of intact allergens were reported for both, nano- as well as micro particles [108,109,110]. Application of PLGA particles loaded with bee venom phospholipase A2 for SIT was shown to protect mice from severe allergic reactions in bee venom-challenged mice. Desensitization with microparticles loaded with phospholipase A2 (PLA2) and protamine stabilized CpG elicited a Th1 type response associated with higher IgG2a titers compared to generic Alum mediated therapy. The high IgG2a titers were correlated to a reduced anaphylactic reaction in the form of attenuated rise in body temperature upon challenge with bee venom [111].

According to the literature availability, PLGA particles provide a promising tool for immunotherapy of allergic disease. The applications of PLGA particle mediated immunotherapy in several different disease models of allergy are summarized in Table 2. It is worth mentioning that small clinical pilot studies in humans for PLGA microparticle-mediated immunotherapy for the treatment of grass-pollen allergy have been conducted [112,113]. However, despite favorable properties of PLGA particles, especially in the application for SIT compared to classical approaches, the progress in research towards broad clinical application remains limited. Future studies need to put the focus on clinical transition and applications in humans to meet the desire for novel technologies in the treatment of allergic diseases.

### 3.2. Autoimmunity

Similar to allergy, autoimmune diseases (AID) are the result of the induction of an inappropriate immune response. However, the response is not directed to a foreign antigen that infiltrated the system but against self-antigens leading to the attack and progressing degeneration of certain compartments in the body. The factors involved in AID vary greatly dependent on the affected organs. Generally, the initiation of AID is considered to involve a combination of genetic predisposition and environmental factors [114]. Further, immune imbalances are a common hallmark of AID. Decreased numbers of Tregs and insufficient Treg effector function is reported in several different diseases. This contributes to the development of autoimmunity and is identified as one of the main drivers of autoimmune-related diseases [115]. Immunotherapy against AID is focused around immunosuppression, restoring immune homeostasis and reconstitution of peripheral tolerance [116,117]. PLGA particles excel in their role as carriers for immunosuppressive drugs and tolerogenic vaccines (also called inverse vaccines) and are extensively studied for the application as immunotherapeutic tool in a variety of AID [116].

Multiple sclerosis (MS) is an AID characterized by the infiltration of immune cells into the CNS. Autoreactive T cells recognize self-antigens on oligodendrocytes, which leads to successive destruction of the myelin sheath and subsequent damage of the axons [118]. Immunomodulating drugs are common medication in order to attenuate disease progression. PLGA particles are investigated in the context of potential drug carriers for the delivery of immunomodulators to the CNS. Capitalizing on drug protection and controlled release from PLGA particles, interferon (IFN) β-1a has been encapsulated into PEG-PLGA nanoparticles. The advantageous properties of PLGA allowed for reduced dosing compared to standard IFN β treatment and could present itself as a less toxic alternative for MS treatment [119]. The in vitro toxicity study performed by the group was limited to hepatocytes, which play a major role in detoxification and are subjected to potentially toxic metabolites. However, an important goal of immunotherapy for MS treatment is the delivery of therapeutic agents to the CNS. Thus, toxicity studies on the effects of PLGA particles on cells of the CNS should be performed. Trafficking the BBB by PLGA particles has been shown by de la Flor et al. In their study, the group functionalized leukemia inhibitory factor (LIF)-loaded PLGA nanoparticles with anti-CD4 to target CD4 T cells in the CNS. Delivery of LIF to CD4 T cells induced the Treg phenotype in these cells and an anti-inflammatory effect in the frontal cortex of the brain could be shown. The efficacy of the formulation was tested in experimental autoimmune encephalomyelitis (EAE), a mouse model for MS. Daily intraperitoneal administration of PLGA LIFNano-CD4 particles ameliorated clinical scores of EAE significantly and reduced pro-inflammatory IL-6 levels in the frontal cortex to the levels of control mice [120].

A potent strategy to overcome autoimmune responses is the development of tolerogenic vaccines. PLGA particles exhibit an ideal vaccine platform for immunogenic but also tolerogenic vaccines. The possibility to adapt antigen dosing and the co-delivery of immunostimulatory or tolerogenic adjuvants, release kinetics and targeted delivery to APCs facilitates the formulation of tailored vaccines [116]. Common antigens for tolerogenic vaccines against MS include myelin oligodendrocyte glycoprotein (MOG) and proteolipid protein (PLP).

Formulation of the immunodominant epitopes MOG_35–55_ and PLP_139–151_ into PLGA particles was reported to induce tolerogenic responses upon co-delivery with tolerogenic adjuvants in several studies. Subcutaneous injection of both PLGA nanoparticles encapsulating MOG antigen and PLGA nanoparticles encapsulating immunosuppressive IL-10 simultaneously ameliorated clinical scores of EAE in C57BL/6 mice and demonstrated immunosuppressive functionality. T cells that were stimulated in vitro with MOG produced significantly lower amounts of pro-inflammatory cytokines IL-17 and IFN-γ [121]. Cho et al. developed a dual-sized PLGA microparticle system for tolerogenic immunization against MS. MOG_35–55_ was encapsulated into smaller sized microparticles of 1 μm. Co-delivery of the immunosuppressive modulators transforming growth factor (TGF)-β and granulocyte macrophage–colony stimulating factor (GM-CSF) was provided by encapsulating them into larger microparticles of 45–60 μm in diameter. Injection of the microparticles delivered the MOG antigen to DCs via uptake of the small particles in an immunosuppressive environment of the released TGF-β/GM-CSF from the depot of the larger particles. Tolerogenic vaccination inhibited EAE in a semi-therapeutic mouse model. A significant reduction of the clinical score was accompanied by reduced infiltration and secretion of pro-inflammatory cytokines in EAE mice [122]. Covalent conjugation of MOG_35–55_ to PLGA nanoparticles was found to ameliorate EAE in mice when immunized subcutaneously in a prophylactic manner. Additionally, effects on immune homeostasis were reported in the form of increased populations of Tregs in the spleen. Ex vivo stimulation of the splenocytes resulted in lower secretion of pro-inflammatory IL-17 and IFN-γ but in turn increased IL-10 production [123]. A study from 2021 investigated an alternative to the co-delivery of tolerogenic adjuvants. It was shown that PEGylation of PLGA nanoparticles containing MOG peptide was sufficient to induce therapeutically relevant tolerance in a mouse model of MS. While MOG-PLGA-nanoparticles did induce DC maturation and the expression of co-stimulatory markers after subcutaneous injection, PEG-MOG-PLGA-nanoparticles showed significantly reduced DC activation [124]. An approach of peptide cargo modification instead of the particles was reported by Triantafyllakou et al. In their study, MOG_35–55_ was conjugated to glucosamine which facilitates the internalization by DCs via interaction with the mannose receptors. Encapsulation of the glucosamine–peptide conjugate into PLGA nanoparticles achieved an induction of a tolerogenic response in mice, effective in the prevention and the treatment of EAE [125]. This may circumvent the obligatory addition of adjuvants in tolerogenic vaccines to further improve the safety profile.

Next to MOG_35–55_, the immunodominant epitope of the proteolipid protein (PLP_139–151_) was contemplated by researchers as a target for the development of PLGA particle mediated tolerogenic vaccines in MS. Lima et al. tested the encapsulation of PLP_139–151_ into PLGA nanoparticles that were subsequently incorporated into poly(vinyl alcohol)-poly(vinyl pyrrolidone) microneedles. These polymeric microneedles are designed for lower invasive administrations into the skin. Characterization of the PLGA particles and the microneedles revealed efficient encapsulation and release under physiological conditions [126]. However, no in vivo experiments or biologic activity assays have been performed in this study. The efficacy of this novel approach for a tolerogenic vaccine needs to be further investigated in order to optimize the formulation and clarify whether this form of administration can omit the use of tolerogenic adjuvants. Interestingly, the route of administration seemingly affects the intrinsic adjuvant activity of PLGA particles. While subcutaneous injections of PLGA particles as inverse vaccines needed the addition of immunosuppressive adjuvants such as IL-10 or TGF-β to induce effective tolerogenic responses in mouse models of EAE, intravenous administration of PLGA particles encapsulating solely the antigens were sufficient to ameliorate the disease in mice [127,128]. Here, the dosing needs to be considered. Intra venous administration of a high dose of nanoparticles, chemically conjugated with a high amount of PLP_139–151_ ameliorated EAE in mice to a higher extent than lower dosing of the vaccine or the antigen. Investigation of in vitro DC maturation revealed a decreased expression of co-stimulatory markers and an inversely proportional increase in antigen presentation with increasing peptide concentrations added to the medium [129]. In a study by Elahi and co-workers, PLGA particles were investigated for therapy of Guillain–Barré Syndrome (GBS). Similar to MS, GBS is an AID associated with inflammation and demyelination of nerve cells. However, in this case, predominantly neurons of the peripheral nervous system are affected [130]. Interestingly, cargo-free PLGA particles used in this study accomplished disease amelioration in a murine model of GBS compared to peptide immunized controls with fewer monocytes being found in the blood and the spleen. Thereby, expression of pro inflammatory markers for IL-17, IFN-γ, and TNF-α were reported to be significantly reduced on an mRNA level [131]. The authors attribute the observed effects to the adhesion of circulating monocytes to the PLGA NP, deterring them from infiltrating the inflammatory sites. This thesis should further be mechanistically investigated.

PLGA particle-based inverse vaccines have proven to be effective in other AID as well, one of which is type I diabetes (T1D). T1D is a chronic AID that involves the progressive destruction of the insulin-producing β-cells in the islet of the pancreas by autoreactive immune cells. Especially the infiltration of autoreactive T cells plays a crucial role in the immunopathology of T1D but also the production of antibodies reactive to, e.g., insulin have been reported [132]. A dual-sized PLGA microparticle system was developed by Lewis and colleagues to induce a tolerogenic response to an insulin peptide. Therefore, unphagocytosable microparticles of a diameter of 30 μm have been fabricated encapsulating either TGF-β or GM-CSF. These generated an immunosuppressive milieu at the injection site that was supported by phagocytosable smaller PLGA microparticles encapsulating Vitamin D3. The injection of these immunosuppressive microparticles together with PLGA microparticles that encapsulated the epitope insulin B_9–23_ significantly delayed the onset of T1D in NOD (non-obese diabetic) mice, a mouse model for the spontaneous development of T1D. Further, the tolerogenic microparticle vaccine could significantly reduce the insulitis score in the islet of the pancreas and increase the number of immunosuppressive Tregs and GR-1^+^ DCs and macrophages in the spleen [133]. Another two-component system was developed for tolerogenic immunizations against insulin by Yoon et al. The peptide hydrogel PuraMatrix^TM^ containing GM-CSF and CpG was introduced as an adjuvant for the recruitment of immune cells to the insulin baring PLGA microparticles upon co-administration of both delivery systems. The inverse vaccine delayed the onset of T1D in NOD mice significantly and prevented the disease in 40% of the animals [134]. In fact, microencapsulation of insulin B_9–23_ into PLGA particles for inverse vaccination as a therapy for T1D alone was shown to be effective in NOD mice. Weekly vaccinations with insulin B microparticles over five weeks prevented the onset of T1D and significantly protected β-cell islet destruction in pancreas [135].

Rheumatoid arthritis (RA) presents another AID as a target for a potential tolerogenic vaccine. The disease is associated with infiltration of autoreactive T cells, B cells, and macrophages into the synovial membrane in different joints. Activation of endothelial cells and subsequent swelling and secretion of pro inflammatory cytokines promotes the progressive joint damage. The cause of RA is still not known. However, potential autoantigens have been identified as drivers of disease progression and are shared among many patients such as type II collagen (CII), N-acetylglucosamine-6-sulfatase and filamin A [136,137,138]. The attempts of researchers to formulate a tolerogenic PLGA particle-based vaccine for the treatment of RA have been met with success. In a mouse model of collagen-induced arthritis, oral pretreatment of mice with PLGA nanoparticles loaded with CII could prevent the onset of RA. In a therapeutic setting, oral treatment with CII nanoparticles achieved significant suppression of the disease progression compared to control animals. When analyzing the draining lymph nodes and blood serum, lower titers of CII-specific IgG antibodies and secretion of pro-inflammatory TNF-α was detected, but the levels of TGF-β were found to be increased [139]. In a study in rats, a single intra muscular vaccination with PLGA microparticles containing an altered CII peptide was sufficient to inhibit the development of collagen-induced arthritis with comparable therapeutic success as multiple intravenous injections over three weeks [140], highlighting the potential and the benefits of PLGA particle-mediated inverse vaccines in dose reduction and delivering the substances in a less invasive way. Other than inducing tolerogenic immune responses with inverse vaccines, general immunotherapy for AID and RA aims at the targeted delivery of anti-inflammatory agents for immunosuppression of autoreactive immune cells. With the multitude of potential drugs and modifications that can be incorporated into or conjugated or adhered to PLGA particles, the possibilities for PLGA application in this field are vast. In a study from 2020, Haycook et al. utilized the potential of PLGA encapsulating the inhibitor eggmanone into nanoparticles and conjugated CD4 F(ab’) antibody fragments to the surface of the particles. The highly efficient targeted delivery of eggmanone to CD4^+^ T cells did impair the effector function of these cells in vitro suggesting promising future applications to silence autoreactive T cells in AID such as RA [78]. Another eminent example for targeted delivery of an immunosuppressive drug to sites of inflammation in RA is presented in the study by Li et al. Encapsulation of tacrolimus, a calcineurin inhibitor, into PLGA nanoparticles and subsequent coating of the particle surface with membrane microvesicles (T-MNP) originated from a RAW 264.7 macrophage cell line accomplished efficient drug delivery to the inflamed joints via “mimicking” macrophages which are recruited to sites of inflammation via their surface chemokine and cytokine receptors. When applied to a mouse model of collagen-induced arthritis, T-MNP treatment inhibited disease development significantly and almost abolished inflammation in the joints of the mice [141]. Interestingly, membrane-coating of drug-free PLGA particles alone was found to be effective in the treatment of RA. PLGA nanoparticles, coated with neutrophil-derived membranes, were sufficient to reduce synovial infections and clinical scores in two mouse models of RA. The study showed that neutrophil-membrane-coated PLGA nanoparticles neutralized pro inflammatory TNF-α and IL1-β, reducing their levels in the blood plasma and in the joints [142].

Delivery of immunomodulatory drugs via PLGA particles for the treatment of AID has been extensively studied in various diseases including psoriasis [143], dermatitis [144], and systemic lupus erythematosus (SLE). The latter is an autoimmune disease attributed to autoreactive B and T cell activation, resulting in multiple-organ damage mainly caused by autoreactive antibodies and Th17-mediated secretion of pro-inflammatory cytokines [145]. PLGA particles have been applied in recent SLE research to reconstitute Treg induction and immune homeostasis. PEGylated PLGA-poly(L-lysine) nanoparticles have been fabricated in a study by Zhang et al. Loading of these particles with microRNA(miRNA)-125a protects the otherwise fragile RNA molecule from degradation and allows for efficient delivery to T cells in the periphery. Intravenous injection of PEGylated PLGA-PLL particles loaded with miRNA-125a ameliorated disease symptoms and progression to a comparable amount as a standard Dexamethasone treatment. Thereby, no unspecific depletion of lymphocytes in the periphery but an increase in the Treg population was observed. The increase in Tregs re-established immune homeostasis and mitigated T cell effector functions depicted by reduced IFN-γ and STAT3 (signal transducer and activator of transcription 3) mRNA levels in splenic T cells [146]. Induction of increased Treg populations to restore immune homeostasis in SLE was also aimed at by Horwitz and colleagues. In their study, TGF-β was encapsulated together with IL-2 into PLGA nanoparticles. The particles were subsequently coated with anti- CD2/CD4 antibodies to target the nanoparticles to CD4^+^ and CD8^+^ T cells and induce Treg expansion. The consequently to PLGA particle administration expanded Treg cells accomplished disease and symptom amelioration in a mouse model of SLE [147].

In summary, PLGA particles present an ideal platform for immunotherapy of AID (Table 3). Antigens associated with autoimmunity can be efficiently co-delivered with tolerogenic adjuvants in a controlled manner. Further, side directed drug delivery via surface modifications of PLGA particles even across physiological barriers enables targeted therapy on a cellular level, minimizing adverse side effects and drug dosing.

### 3.3. Infectious Diseases

Outbreaks of infectious diseases throughout the ages have posed a constant threat to the human population. Advances in medicine and milestone discoveries such as antibiotics and the efficacy of vaccines in protection from infections enabled humanity to overcome the by then high morbidity and mortality rates of diseases such as tuberculosis, polio, diphtheria, bubonic plague, and smallpox. In modern societies with ever progressing globalization and global trafficking, new challenges in containing the emerge and spread of infectious diseases arise. Recent epidemic outbreaks of Ebola, Zika virus disease, and the global COVID-19 pandemic are alarming examples of the imminent threat of infectious diseases. Beyond that, diseases such as HIV, tuberculosis, or malaria are still abundant in the human population [148]. This conveys the need for novel options to treat and prevent infectious diseases and impede the epidemic spread in the population. PLGA particle mediated immunotherapy represents a promising tool with diverse scope for potential applications. Targeted delivery of antibiotics loaded to PLGA particles has been described in several studies to effectively treat bacterial infections and reduce the risk of resistance development [149,150,151]. In a similar fashion, encapsulation and delivery of anti-viral and immunomodulating drugs have been successfully explored using PLGA particles to fight viral infections [152,153,154,155]. Aside from the aforementioned tools for approaching infectious diseases, protective vaccinations represent one of the most effective medical tools up to date. Most of the currently applied vaccines are aimed at inducing a humoral response against a certain pathogen. The most commonly used adjuvants in that regard are aluminum salts. Many aluminum salt-based vaccines are approved for the use in humans and excel at inducing a Th2 type response with the generation of long-lived antibody responses. However, their efficacy against intracellular pathogens remains limited due to the lack of a potent cellular response [156]. To overcome the limitations of current vaccines and expand the superb efficacy in containing infections to other infectious diseases, the development of next generation vaccines is inevitable. PLGA particles as vaccine platforms have been extensively studied, yielding promising results for the development of PLGA-based next-generation vaccines [157]. The tailorable nature of PLGA in terms of physicochemical properties and the protection of the cargo from degradation sets almost no limits to the encapsulation of diverse antigen sources and immunomodulatory adjuvants into PLGA particles. Further, particle modification enables targeted delivery to specific tissues or immune subsets, facilitating a desired immune response optimized for the protection from specific pathogens [158].

The rationale behind vaccines is the delivery of antigens to APCs, most importantly dendritic cells. The antigen needs to be processed and presented on the cell surface on MHC class II and cross-presented on MHC class I molecules. Alongside, co-stimulatory signals need to be provided in order to prime naïve T cells and induce an adaptive immune response against the antigen of choice. Therefore, cellular uptake of PLGA based vaccines and subsequent maturation of DCs is the main goal of PLGA mediated vaccine development. Encapsulation of antigens into PLGA particles is sufficient to induce a specific immune response in mice, outlining the intrinsic immunogenicity of PLGA particles [159,160,161]. However, no upregulation of maturation markers on human monocyte derived DCs upon uptake of PLGA microparticles in vitro was observed in other studies [162]. To ensure DC maturation, which is mandatory for eliciting a potent immune response, and avoid the risk of antigenic tolerance induction, immunostimulatory adjuvants are commonly incorporated in PLGA particulate vaccine formulations [40,163,164].

Viral hepatitis infections by hepatitis B (HBV) and C (HCV) virus represent the major course of chronic viral hepatitis worldwide leading to high rates of liver disease-mediated premature death [165]. As a global health problem, many efforts on the development of treatment options and prophylactic vaccines for viral hepatitis have been made. Chong et al. formulated a PLGA nanoparticle vaccine encapsulating the HBV nucleocapsid core antigen together with the TLR4 ligand monophosphoryl-lipid A (MPLA). The authors found a robust induction of a Th1 type immune response upon vaccination of mice with the PLGA vaccine that was superior to soluble vaccination. The measured IFN-γ response could further be increased after a booster immunization 14 days after the prime [166]. In a study by Thomas et al., HBV surface antigen (HBsAg)-loaded nanoparticles were administered to rats via aerosols to the mucosa of the airways. The inhalable particles elicited robust cellular and humoral responses specific to HBV. Of particular interest is the generation of an IgA antibody response, which could prevent infections with HBV through mucosal exposure. Indeed, IgA antibodies were found in the bronchoalveolar lavage as well as in the vaginal lavage [167]. Investigations on targeted delivery of HBsAg to dendritic cells were performed in a study on mannose-modified PLGA nanoparticles. The mannose surface modification of the PLGA vaccine accomplished targeted delivery to APCs and increased internalization via interaction with the mannose-receptors on APC surfaces shown in vitro. Upon vaccination of mice, the targeted vaccine did not alter the effectiveness of the induced humoral response but significantly boosted the elicited cellular response [168]. Just recently, a particulate PLGA-based vaccine for hepatitis was formulated and investigated in vitro [169], emphasizing the need but also the potential of particulate vaccines in combating viral hepatitis diseases. This is especially true for HCV as to date no vaccine has entered the market. Hekmat et al. developed a PLGA-based nano-conjugated vaccine delivery system utilizing a recombinant core-NS3 fusion protein. Characteristics of the particles have been assessed qualifying them for the use as an HCV specific nano-vaccine. However, no biologic tests have been performed in the study to confirm the efficacy of the vaccine [170]. Roopngam and co-workers formulated a PLGA microparticle vaccine encapsulating the E2 envelope glycoprotein of HCV. In mice, immunized with the microparticulate vaccine, significantly higher CD8^+^ T cells responses against E2 compared to soluble immunizations were observed. Increased levels of ex vivo-secreted IFN-γ were measured upon PLGA particle immunization as well as a robust IgG antibody response [171].

Influenza depicts an acute viral infection of the respiratory tract mainly caused by the influenza A (IAV) and B (IBV) virus. Seasonal flu outbreaks are common events in most countries and are accompanied with several hundred thousand deaths per year. In addition, epidemic or even pandemic outbreaks such as the Spanish flu in 1918 are an eminent threat to the human population. Especially IAV plays an important role in these outbreaks [172]. IAV is subjected to constant alteration of in particular the surface antigens hemagglutinin (HA) and neuraminidase (NA), attributed to antigenic drift and antigenic shift. Antigenic drift are changes in the HA or NA due to point mutations in the fragmented RNA genome of IAV and is a frequently occurring event. Antigenic shift describes an exchange of HA or NA genome fragments with other virus strains. This leads to the generation of completely new strains of IAV. Though antigenic shift is a very rare event, it is involved in the dynamics of epidemic outbreaks of influenza [173]. Hence, annual vaccination and the adjustment of the vaccines to the dominant strains is a crucial instrument in controlling influenza infections. PLGA particles have been explored as a vaccine platform for IAV in multiple studies. Zhang et al. prepared PLGA microparticles encapsulated with the TLR7 ligand imiquimod and loaded with HA from an H5N1 variant split vaccine. Bone marrow-derived DCs (BMDCs) and peritoneal macrophages, treated in vitro with PLGA microparticles demonstrated maturation by the secretion of pro inflammatory cytokines. Analysis of the immune response in mice after vaccination with the PLGA particles revealed a strong antigen-specific antibody response and an induction of a Th1 biased cellular immune response [174]. PLGA nanoparticles have been used as an influenza vaccine system in a study in chickens. Inactivated antigens of the avian influenza strain H4N6 were encapsulated into PLGA particles with or without CpG as adjuvant. Adjuvanted particles exhibited strong IgM and IgG antibody responses in the serum and responded to in vitro stimulation with influenza-specific secretion of IFN-γ. Interestingly, modification of the PLGA nanoparticles with chitosan enabled mucosal immunization of chicken via aerosols. In addition to IgM and IgG immunoglobulins, specific IgA was detected in the mucosa of the airways upon chitosan-PLGA particle vaccination in aerosols [175]. An additional virus challenge to test the potential of the PLGA particulate vaccine in containing and/or preventing an influenza infection should be performed in future studies. Although the flu vaccines are adjusted annually, immune escape driven by antigenic shift and antigenic drift implicate a major problem in influenza vaccine development. Frequent adjustment of the formulation requires high efforts and funds. Further, prediction of the dominant variants for the seasonal flu is often difficult. To overcome these limitations of vaccines directed against the IAV surface antigens, we and others have developed PLGA particle-based vaccines for the immunization against conserved IAV antigens [89,176]. Seth and colleagues encapsulated a recombinant capsomere presenting M2e peptide antigen of the IAV matrix protein 2 into PLGA particles. A mainly IgG1-driven antibody response as well as a cellular response with M2e specific cytotoxic T lymphocytes (CTLs) were detected in mice immunized with the PLGA particles [176]. In our lab, Herrmann et al. encapsulated the HLA-A*0201-restricted epitopes of the IAV M1 matrix protein and the PA subunit of the polymerase complex into PLGA microparticles. Immunization of humanized transgenic mice elicited considerable IAV-specific CTL responses. Subcutaneous prime and intra nasal boost vaccination of mice with PLGA microparticles protected mice from an IAV challenge as a consequence of CTL responses in the airways [89]. In a follow-up study, MacKerracher and co-workers analyzed the induction of T cell memory after PLGA microparticle immunizations with a focus on tissue resident memory cells (Trms) in the lung. A large population of CD8^+^ Trms were detected in the lungs and the bronchoalveolar lavage fluid consequently to subcutaneous prime/intra nasal boost vaccinations. The abundance of these specific Trms was sufficient to protect mice from a challenge with recombinant IAV [177]. This study emphasizes the potential of PLGA particles as next-generation vaccines with tailorable properties to elicit humoral as well as strong cellular immune responses and induce memory formation.

The importance of vaccines’ ability to induce a robust cellular immunity in addition to humoral responses becomes clear in the vaccine development for the human immunodeficiency virus (HIV) [178]. HIV infects immune cells expressing the CD4 receptor, e.g., CD4^+^ T cells and DCs, which leads to a progressive degeneration of the immune system and eventually results in the onset of acquired immunodeficiency syndrome (AIDS). Binding to CD4 and the chemokine receptors CCR5 or CXCR4 initiates cellular infection. HIV is a retro virus able of transcribing its RNA genome into DNA. Upon infection, the reversely transcribed DNA can be incorporated into the host cells’ genome utilizing the viral integrase. That way, the viral genome can persist within the host cells protected from clearance by the immune system [179]. Despite exceptional progress in the treatment of HIV by highly active antiretroviral therapy (HAART), no cure has been found to date. At the same time, no effective vaccine formulations for the prevention of HIV infections have been accomplished yet. PLGA particulate vaccines have the favorable properties of eliciting Th1 biased cellular immune responses, which are needed to effectively identify and eliminate infected cells. Hence, PLGA particles exhibit an ideal delivery system for HIV vaccine development. Indeed, a PLGA microparticle system encapsulating the HIV antigen gp120 has been formulated for vaccination against HIV. Adjuvanted with SQ21, the PLGA vaccine induced a neutralizing antibody response in guinea pigs mediated by an “autoboost” after bulk particle erosion [180]. In a different study, encapsulation of a PEI/DNA complex into PLGA microparticles was investigated as a gene-based vaccine against HIV. The plasmid DNA encapsulated into the PLGA particles encoded for several viral genes, i.e., the Gag, Pol and Env genes. Three immunizations of BALB/c mice over 8 weeks elicited HIV-specific antibody and potent CTL responses, which were conformed for their cytolytic potential in an ex vivo cytotoxicity assay [181]. Despite encouraging results, further research on PLGA particle-mediated vaccines is mandatory to capitalize on its favorable properties as vaccine platform for HIV.

Lastly, the PLGA particles have been examined for the use as carriers in the development of vaccines against severe acute respiratory syndrome-coronavirus-2 (SARS-CoV-2). Vaccines are key to overcome the COVID-19 pandemic and foster immunity against SARS-CoV-2 in the population. Current vaccines approved for the use in humans are effective in generating humoral and cellular immune responses in order to prevent infections and severe cases of COVID. However, the development of next-generation vaccines is mandatory for mucosal applications and long-lasting memory induction. Reports on PLGA particles utilized for effective vaccine delivery show strong inductions of T cell and B cell immunity against the spike protein and envelope protein of SARS-CoV-2 [182]. In a trivalent vaccination strategy of subunit vaccines against SARS-CoV-2, PLGA implants were formulated containing the trivalent subunits for sustained release. Compared to the other delivery strategies of the study, one administration of PLGA-mediated vaccine delivery was sufficient to induce Th1 biased T cell responses and neutralizing antibody titers in mice [183].

At least since the COVID-19 pandemic, the need for efficient next-generation vaccine platforms in order to control or even prevent future epidemic outbreaks of infectious diseases became clear. The potential pioneer role of PLGA-based particles in vaccine development is emphasized by the recent advances and increasingly high-impact publications in the field that have been summarized in Table 4. The properties of PLGA particles to induce antigen-specific Th1 type cellular immunity in addition to long-lasting antibody responses display a distinct advantage in comparison with classical vaccine platforms. Beyond that, studies on PLGA particle-based vaccines have demonstrated safe and efficient mucosal immunizations with IgA antibody and mucosal T cell responses which is needed to achieve sterile immunity [184].

### 3.4. Cancer Immunotherapy

Cancer represents one of the leading malignancies and causes of death worldwide. Standard treatment options encompass radical surgery, radiation therapy and especially in progressive states of the disease chemotherapy. PLGA particles are applied in the clinics for chemotherapeutic cancer treatment accomplishing efficient drug delivery and a reduction in adverse side effects with more PLGA particulate formulations in clinical trials [185,186,187]. However, patients suffer from devastating side effects of classical cancer treatment due to unspecific destruction of healthy tissue alongside the cancer cells. Further, tumor relapse and metastasis formation are often inevitable as a consequence of resistance development to the treatment [188]. With emerging success, anti-tumor immunotherapy has shifted into the focus of researchers worldwide. Various strategies for the modulation and mobilization of the immune system to recognize and eliminate cancer cells have been developed and are distinguished into two broad categories: Passive immunotherapy which implies, e.g., monoclonal antibodies (mAb) targeting immune checkpoint inhibitory receptors or tumor markers and active immunotherapy, which aims at eliciting a tumor-specific immune response [189]. Transition to the clinics resulted in spectacular results and especially immune checkpoint inhibition with mAbs is considered a standard therapy option for various malignancies [190].

PLGA particles have been investigated as delivery platforms for immunotherapy. The potential for targeted delivery via surface modifications, tailorable drug release profiles, and protection of the cargo from enzymatic degradation predestines PLGA particles as ideal systems for immunotherapy. Conjugation of anti-PD-L1 F(ab) fragments to PEGylated PLGA nanoparticles was shown to improve the efficacy of immune checkpoint inhibition compared to soluble administration of anti-PD-L1 mAbs. In addition to prolonged drug circulation in mice, intra peritoneal injections of anti-PD-L1-PEG-PLGA particles resulted in higher frequencies of splenic CD8^+^ T cells and B cells as well as higher infiltration of macrophages, neutrophils, and natural killer (NK) cells into the tumor. In a subcutaneous model of MC38 colon cancer, PLGA particle therapy was similarly efficient as the treatment with soluble anti-PD-L1 mAb, resulting in significantly reduced tumor growth [191]. In another study, multifunctional PLGA nanoparticles were fabricated by encapsulation of anti-PD1 mAbs together with iron oxide and perfluoro pentane. The particles were surface-modified with PEG and GRGDS peptide sequences for directed delivery into melanoma tumor tissue. The combination with photothermal therapy (PTT) and site-directed drug delivery mediated by PLGA particles accomplished enhanced anti-tumor activity via effective anti-PD1 mAb release and infiltration of CD8^+^ T cells into the tumor. This combination therapy significantly prolonged tumor-free survival in a mouse model of solid melanoma tumors [192]. The application of PLGA particles as delivery systems of immune checkpoint inhibitors represent a promising approach to achieve a dose reduction which may prevent induction of autoimmunity often accompanied with immune checkpoint inhibition therapy.

One challenge in cancer immunity is dealing with the tumor microenvironment. With tumor progression, cancer cells adapt to immune infiltration and generate an immunosuppressive microenvironment surrounding the tumor tissue. In these “cold” tumors, overall immune cell infiltration is significantly reduced, and infiltrating effector cells are silenced by a milieu of immunosuppressive cytokines, abundant Tregs, and myeloid-derived suppressor cells (MDSC). One approach in cancer immunotherapy is to overcome this mechanism of immune evasion by tumors and transform the microenvironment towards a pro-inflammatory milieu and re-establish the immunogenicity of “hot” tumors [193]. Encapsulation of certain chemotherapeutics into PLGA nanoparticles was found to induce immunogenic cell death in pre-established tumors leading to the release of damage-associated molecular patterns (DAMPs). DAMPs have been shown to activate immune cells and are thus contributors to immunomodulation of the tumor microenvironment [194]. In vitro studies revealed a significantly stronger induction of DAMP-mediated DC and T cell activation in response to PLGA particle-induced immunogenic cell death compared to soluble cytostatics. In a mouse model for pancreatic cancer, PLGA particle treatment significantly reduced the tumor burden. This effect was to a large margin attributed to immunotherapeutic effects as it was less prominent in immunodeficient mice. Interestingly, the effect was negligible in mice treated with soluble cytostatics [195]. Modulation of the tumor microenvironment in another study was obtained by PLGA particulate delivery of metformin-CO_2_ to tumor-associated macrophages (TAMs). The PLGA nanoparticles were camouflaged by coating with a hybrid biomimetic membrane originated from cancer cells and macrophages for targeted delivery. pH-dependent release of CO_2_ from encapsulated metformin co-delivered with an siRNA for silencing FGL1 expression—the ligand for lymphocyte-activation gene 3 (LAG3), a known immune checkpoint receptor—reduced the immunosuppressive nature of the tumor microenvironment and improved the tumor-free survival in a mouse model for triple negative breast cancer significantly [196]. A similar approach of camouflaged PLGA nanoparticle delivery to TAMs was pursued by Zhang et al. PLGA nanoparticles loaded with the TLR7/8 agonist R848 were coated with tumor cell-derived membranes and targeted to M2 like TAMs via PEP-conjugation to M2pep. Targeted delivery of R848 to the M2 type TAMs induced a reprogramming into the pro-inflammatory M1 type in these cells. The subsequent pro-inflammatory modulation of the microenvironment allowed for the induction of a tumor specific immune response, infiltration of effector CD8^+^ T cells, and tumor eradication in a B16/F10 melanoma mouse model [197]. In general, PLGA particle delivery of TLR ligands to modulate the tumor microenvironment has been reported in several studies to be an effective tool for immunotherapy [198,199,200]. In a study of Hao and co-workers, a bi-modular PLGA-based delivery system was applied. The chemokine CXCL-1, a chemoattractant for neutrophils, was loaded into a PLGA hydrogel. Further, the chemotherapeutic agent paclitaxel was encapsulated into PLGA nanoparticles. Intravenous application of paclitaxel-loaded PLGA nanoparticles reached the neutrophils in the circulation. The pre-inoculation of CXLC-1 loaded PLGA hydrogel near the tumor site created an artificial tumor microenvironment by the sustained release of CXCL-1 that attracted paclitaxel-nanoparticle bearing neutrophils to the tumor site in a “trojan-horse” manner. The therapy significantly reduced tumor growth in a mouse model of B16 melanoma [201].

In active cancer immunotherapy, special emphasis has been placed on PLGA particle-mediated cancer vaccines. To accomplish eradication of solid tumors, a robust Th1 type cellular response with potent CTL induction is mandatory. Vaccine formulations with PLGA particles as carrier polymers generally elicit Th1 biased cellular responses in contrast to standard adjuvants such as alum salt formulations which tend to stimulate Th2 type responses with a focus on humoral responses rather than cellular immunity [202]. Therefore, great efforts have been made to develop effective cancer vaccines based on PLGA delivery systems. Besides the appropriate delivery system, efficient cancer vaccines are dependent on two other essential components. Firstly, immunogenic tumor-associated antigens (TAA) have to be identified and provided and secondly, strong Th1 polarizing immunostimulatory adjuvants are necessary to overcome peripheral tolerance to self-derived TAAs and to induce vigorous tumor-specific T cell responses [203].

Various antigen sources have been reported for encapsulation into PLGA particulate cancer vaccines. The most common TAAs used are cancer peptides or epitopes, proteins, and tumor lysates [18,204,205,206]. Delivery of mRNA antigens by PLGA particles was investigated by Yang et al. In their study, PLGA-core/lipid-shell hybrid particles were fabricated. The mRNA antigen was loaded into the lipid shell whereas the TLR7 agonist gardiquimod was encapsulated into the PLGA core as adjuvant. Co-delivery of mRNA and adjuvant to the same APCs in core/shell particles stimulated an antigen specific immune response generating a robust CTL response that did significantly reduce B16 tumor growth in mice [207]. An interesting approach for the formulation of a PLGA nanoparticle cancer vaccine was centered around particle surface modification. Imiquimod-loaded PLGA nanoparticles were coated with cancer cell membranes and further modified with a mannose moiety. The surface proteins on the cancer cell derived membrane did function as the antigen source of the cancer vaccine while the mannose modification improved the targeting to APCs. Targeted delivery of membrane antigens adjuvanted with imiquimod did induce DC maturation in vitro and promoted the secretion of TNF-α and IL-12. Prophylactic immunization of mice in a B16 melanoma mouse model significantly obstructed tumor growth. The effect could be synergistically enhanced when combined with anti-programmed cell death protein 1 (PD-1) mAb mediated immune checkpoint inhibition, demonstrated in a therapeutic B16 melanoma experiment [208]. Surface modification of PLGA microparticles encapsulating tumor lysate with streptavidin/biotin was performed in a study by Gross et al. The improved immunogenicity by coating with streptavidin/biotin resulted in significantly improved protection from metastasis formation in a triple negative breast cancer mouse model for metastasis development [209]. Chen et al. developed a vaccine-like PLGA nano-formulation encapsulating imiquimod together with indocyanine green (ICG). ICG was delivered to the primary tumor in PLGA particles and used for near-infrared laser-triggered tumor elimination. The thereby-generated TAA served as antigen source for the stimulation of an immune response, adjuvanted by the co-delivered imiquimod. The therapy was shown to be effective in two metastases mouse models of breast cancer. The efficacy could further be potentiated by synergistic immune checkpoint inhibition with anti-cytotoxic T-lymphocyte-associated protein 4 (CTLA4) mAbs. In addition, long-term protection by T cell memory formation was reported in the study [210]. In our lab, we co-encapsulated TAAs together with the novel dsRNA analog Riboxxim, a potent agonist for TLR3 and retinoic acid inducible gene I (RIG I), into PLGA nano- and microparticles. Uptake of Riboxxim-loaded microparticles induced maturation of human and murine DC populations in vitro demonstrated by an upregulation of co-stimulatory surface markers and type-I IFN secretion. Subcutaneous immunization of mice with PLGA particles elicited long-lasting antigen-specific CTL responses, superior to those of PLGA vaccines adjuvanted with the classical TLR3 ligand polyIC. In a subcutaneous mouse model of solid thymoma, immunotherapy mediated by PLGA microparticle vaccination reduced tumor growth and prolonged tumor-free survival. Further, additional intra nasal application of the PLGA cancer vaccine stimulated infiltration of specific CTLs into the airways, protecting the mice from metastasis formation in the lung in a B16 melanoma model. Combining the PLGA microparticulate vaccine with anti-CTLA4 mAbs for immune checkpoint inhibition potentiated the therapeutic efficacy of the PLGA particles, accomplishing ablation of pre-established tumors and tumor-free survival in over 50% of the mice. Notably, monotherapy with anti-CTLA4 immune checkpoint inhibition was inefficient, emphasizing the potential of synergistic combination treatment of particulate PLGA cancer vaccines together with immune checkpoint blocking mAbs [84]. Won et al. capitalized on this outstanding synergy of both immunotherapies. In their study, the group approached the immune checkpoint inhibition from a different angle utilizing PLGA particles as a delivery system for siRNA to silence PD-1 and PD-L1 expression in cancer and immune cells. Treatment with PLGA nanoparticles in vitro accomplished a significant reduction in the expression of PD-1 in CD8^+^ T cells and PD-L1 in dendritic cells as well as two different cancer cell lines. Immune checkpoint silencing by PLGA nanoparticles was complemented with a PLGA nanovaccine encapsulating OVA as TAA and polyIC. The double PLGA particle combination therapy strikingly increased tumor-free survival in a subcutaneous mouse model of solid thymoma and initiated tumor ablation in a subcutaneous TC-1 myeloma model [211]. Dölen and co-workers developed a PLGA nanoparticle vaccine encapsulating three peptides of the cancer-testis antigen New York esophageal squamous cell carcinoma-1 (NY-ESO-1) which is known to be widely expressed in several different cancers. The vaccine was adjuvanted by the co-encapsulated α-galactosylceramide (α-GalCer) analog IMM60, a novel agonist for iNKT cells and transactivator for DCs. Immunization of humanized transgenic mice induced strong antigen-specific CD4 and CD8 responses as well as a humoral IgG response. Re-stimulation of NY-ESO-1 sensitized human peripheral blood mononuclear cells (PBMCs) in vitro with PLGA particles induced T cell activation demonstrating the potential for PLGA particle mediated cancer vaccination in humans [212]. Of note, a PLGA nanoparticle formulation of these NY-ESO-1 peptides together with IMM60 is currently investigated in a clinical phase I study [213].

As outlined in this section and in the corresponding summary table (Table 5), a variety of different immunotherapies based on PLGA particles are explored by researchers. Thereby, elegant methods for targeted delivery to the tumor tissue and robust modulation of the immune system and the tumor microenvironment have been developed. These methods were shown to overcome the obstacles associated with cancer therapies such as low immunogenicity and resistance formation. However, malignant cancers are known for rapid adaptations and efficient immune evasion. Therefore, a combination of different treatments is often used in the clinics. Here, PLGA-based immunotherapy has demonstrated striking synergistic effects when combined with additional treatments such as immune checkpoint inhibition. It is now the task to transfer these promising treatment options into the clinics to make PLGA particle-based immunotherapy accessible, especially in malignancies with limited available treatment options.

## 4. Conclusions

PLGA particles exhibit ideal properties for drug delivery in biological systems. Good bioavailability, a supreme safety profile, complete degradation, and non-toxic metabolization of the polymer building blocks enabled FDA and EMA approval for parenteral and mucosal administrations. The biodegradable polymer backbone can protect the loaded substances from premature degradation while simultaneously ensuring controlled and sustained release to the desired organs or sites in the body. As a result, the required drug dose can be greatly reduced due to higher local concentrations of released APIs at the site of action further contributing to the beneficial safety profile. Interactions of PLGA particles with cells of the immune system have been shown in multiple studies making them an attractive drug carrier system for immunotherapy. PLGA particles can be designed for efficient uptake by APCs or targeted to specific cell types via surface modification, thus qualifying them as an ideal drug delivery system for immunotherapy. Delivery of both immunosuppressive and immunostimulatory compounds via PLGA particles for the modulation of inappropriate immune responses has been shown to be effective in a variety of different fields. Encapsulation of immunosuppressive drugs or gene silencers accomplished disease amelioration in allergic and autoimmune diseases. At the same time, delivery of strong immunostimulants in cancer immunotherapy via PLGA particles was shown to modulate the tumor microenvironment, facilitating immune infiltration, and enabling effector functions of infiltrating cells. This shift from cold to hot tumors contributed to tumor ablation and tumor-free survival in serval cancer models.

Particular attention has to be given to vaccine development. PLGA particles excel at the delivery of various types of antigens to APCs and thereby stimulating an antigen-specific immune response. The particles can be designed for efficient uptake and tailored to direct the intracellular trafficking. Co-delivery of encapsulated antigens together with adjuvants was demonstrated to increase antigen presentation on MHC class II as well as cross-presentation on MHC class I by APCs. Co-delivery of tolerogenic adjuvants together with autoimmune-related antigens was associated with Treg driven re-establishment of immune homeostasis and peripheral tolerance in AID. Pairing immunogenic adjuvants with antigens for encapsulation into PLGA particles resulted in the induction of strong Th1 type driven immune responses encompassing humoral and cellular immunity. These responses were reported to be highly efficient in protection from infectious diseases and as immunotherapy for the treatment of cancer. Approaching allergic diseases, both vaccination strategies mediated by PLGA particles were shown to be relevant. Either inverse vaccines for tolerance to allergy associated antigens or modulation of the immunity from Th2 type IgE dominated towards Th1 biased IgG-driven responses was shown to promote desensitization to allergens. In addition, the sustained supply with antigens and adjuvants can circumvent the necessity of several booster applications of vaccines. An overview of vaccinations with PLGA particles is given in Figure 4.

The development of PLGA-based particles for immunotherapy is clearly directed to application in the clinics. Many studies on particulate PLGA systems shifted towards the development of PLGA particle formulations using agents approved for the use in humans and tend to optimize dosing, targeted delivery, and biological safety. Despite major advances in the development of PLGA particle-based immunotherapy, the vast majority of studies are still in a preclinical state. Indeed, the translation into clinical applications of PLGA immunotherapies is accompanied with major challenges. The high demands on drug safety profiles present an issue especially for particles in the nano range, which have to occasionally cope with nanotoxicology. Further, the multiple parameters that influence particle characteristics, although beneficial in designing particles for specific uses, are difficult to control to avoid batch-to-batch variation in regards of the physical-chemical properties. This presents a major challenge particularly in respect of upscaling PLGA particle production. Hence, the setup of standardized manufacturing infrastructures in a GMP environment are mandatory for the clinical translation of PLGA particle-based immunotherapy. However, the successful launch of several PLGA particle-based drugs and their application in the clinics for, e.g., delivery of chemotherapeutics and the consistently effective results in preclinical studies are encouraging for future applications of PLGA particle-based immunotherapy in a variety of immune related disorders and diseases.

In summary, PLGA particle-based immunotherapy represents a promising tool for immunotherapy in allergy, AID, infectious diseases, and cancer. Especially the synergistic therapeutic effects using PLGA particles in combination with established treatments have the potential to significantly improve clinical outcomes and quality of life in patients suffering from a variety of diseases.

## Figures and Tables

**Figure 1 pharmaceutics-15-00615-f001:**
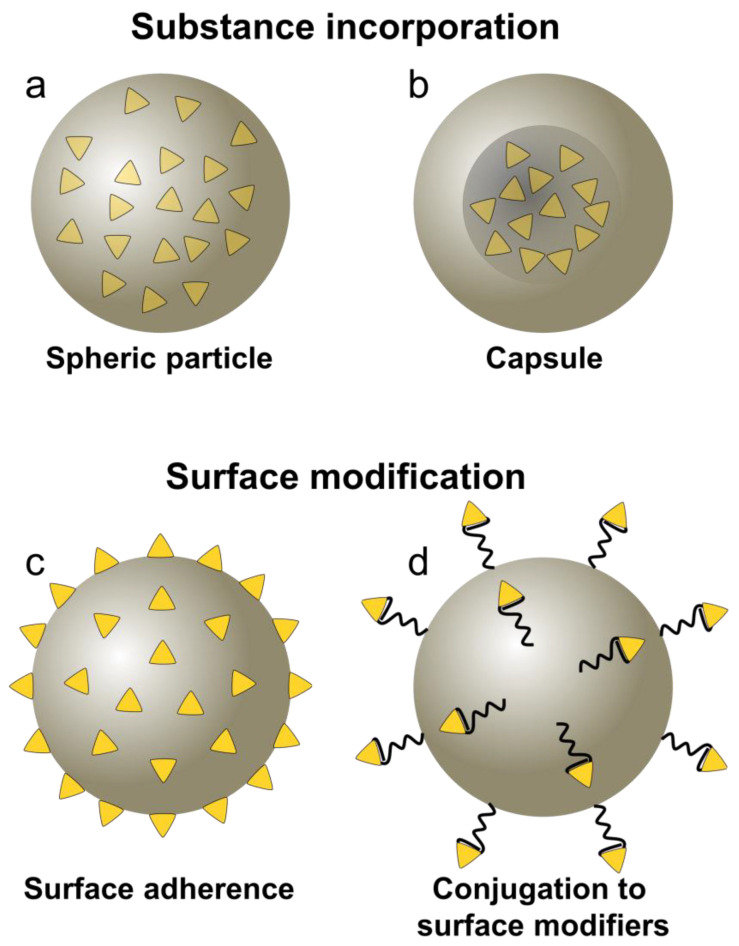
Strategies for particle loading. PLGA particles (gray spheres) can be loaded with substances (yellow triangles) via incorporating them into the particle or loading them onto the particle surface. Substances can be incorporated into PLGA particles via entrapping the molecules into the PLGA matrix of spheric particles (**a**) or encapsulating them into the core of capsules (**b**). Surface loading of cargo molecules can be achieved via adherence to the particle surface (**c**) or by conjugation of the substances to other surface modifiers such as PEG or chemical linkers (**d**).

**Figure 2 pharmaceutics-15-00615-f002:**
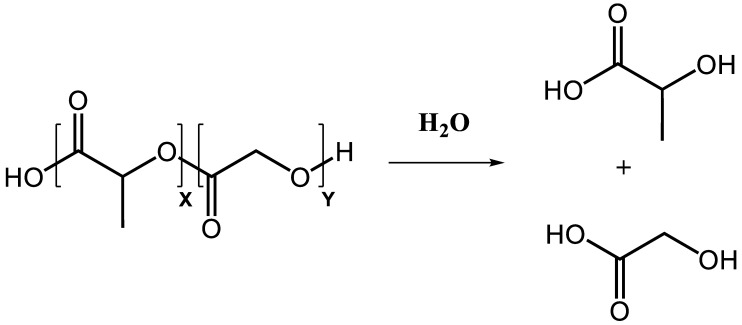
Chemical structure of PLGA and hydrolysis into the monomers lactic acid and glycolic acid.

**Figure 3 pharmaceutics-15-00615-f003:**
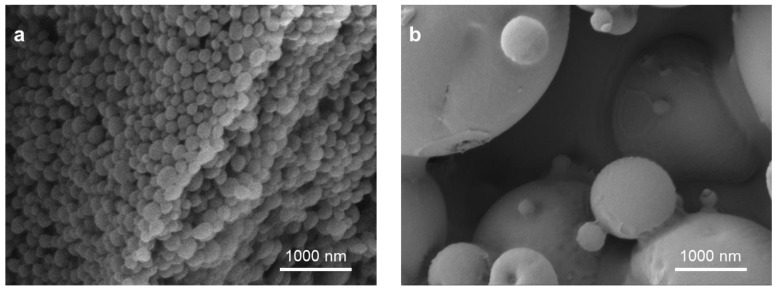
Scanning electron microscopic pictures of (**a**) nano sized and (**b**) micron sized PLGA particles.

**Figure 4 pharmaceutics-15-00615-f004:**
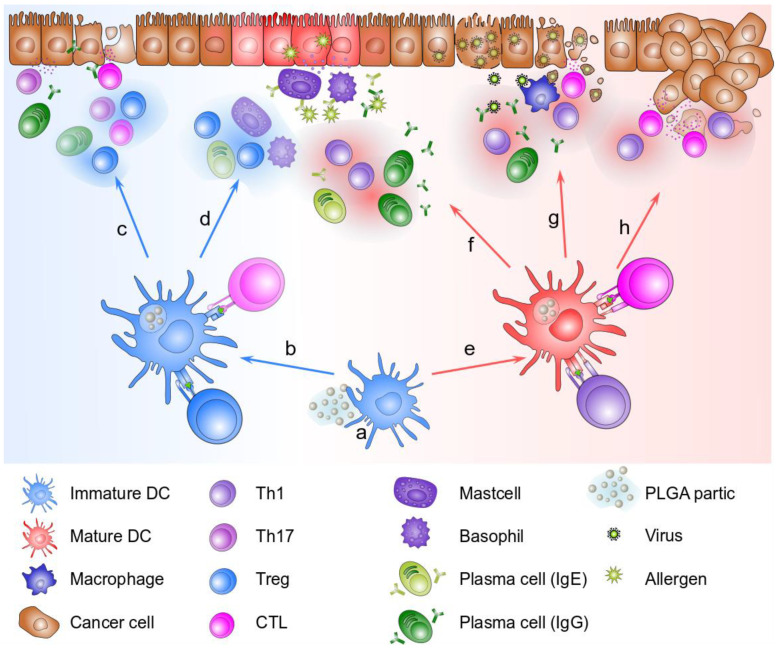
Schematic overview of PLGA particle vaccinations. PLGA vaccines taken up by immature dendritic cells (**a**). Co-delivery of tolerogenic adjuvants leads to antigen presentation on MHC class I and II in the absence of co-stimulatory signals and subsequently promotes anergy in T cells and induces Tregs (**b**). Tregs can silence autoreactive immune responses in AID and re-establish immune homeostasis and peripheral tolerance (**c**). In allergy, Tregs induced by tolerogenic PLGA vaccines can help to ameliorate symptoms via immunosuppression of mast cells, basophils, and effector immune responses such as IgE production by plasma cells (**d**). Upon co-delivery of antigens and immunostimulatory adjuvants, antigen presentation on MHC class I and II, and simultaneous co-stimulatory signaling by DCs to CD4+ and CD8+ T cells elicits potent Th1 biased immune responses (**e**). Th1 mediated SIT induces class switch from IgE to IgG and promotes desensitization in allergic diseases (**f**). Potent protective immunity is achieved by PLGA vaccination resulting in the stimulation of humoral and cellular responses to fight off infectious diseases like viral infections (**g**). PLGA-mediated cancer vaccines can elicit strong Th1 driven cellular immunity capable of tumor eradication by robust CTL responses (**h**).

**Table 1 pharmaceutics-15-00615-t001:** Influences of PLGA particle properties on parameters relevant for immunotherapy.

Parameter	Particle Properties	Influence
Particle distribution	Particle size	PLGA nano-sized particles drain via lymphatic vessels, can be distributed systemically, potential of nanotoxicology PLGA micron size particles reside at site of administration
	Hydrophobicity	Rapid clearance of hydrophobic PLGA particles from the system, unwanted adhesion of particles
	Surface modification	PEG, PEO, poloxamers generate hydrophilic surface PEGylation promotes muco-adhesion in GALT PEGylation protects orally administered PLGA particles from acidic milieu in the stomach to pass gastrointestinal tract
	Route of administration	Intra nasal application of PLGA NP for delivery to CNS
Particle uptake	Particle size	Efficient uptake of PLGA NP/MP by DCs up to a size of 5μm in diameter for antigen presentation and DC activation by cargo molecules
	Surface charge	Negative surface charge of PLGA particles exacerbate particle uptake
	Surface modification	Surface modifications with positively charged polymers to facilitate interactions with negatively charged cell membrane Conjugation/adhesion of molecules targeting phagocytic receptors to facilitate targeted uptake
	Route of administration	Uptake by DCs after intramuscular, intra dermal, and subcutaneous administration Uptake by macrophages after intraperitoneal administration
Immunogenicity	Controlled release	Prolonged antigen presentation
	Intrinsic adjuvant activity	Immune cell activation via NLRP3 activation can circumvent the use of immunostimulatory molecules Risk of unwanted inflammation
	Surface modification	PEGylation enhances interactions with blood components Coating with targeting molecules enables activation of specific immune subsets
	Particle load	Encapsulated antigens are shielded from premature degradation and retain their immunogenicity Accelerated degradation in acidic endosomal compartment facilitates delivery of encapsulated immunomodulators to intracellular receptors
	Route of administration	Route of administration determines Th polarization and efficacy of elicited immune response Intra nasal application of PLGA particles modulates site-directed immune response in bronchoalveolar tract

**Table 2 pharmaceutics-15-00615-t002:** Summary of PLGA mediated immunotherapy of allergic diseases.

Disease	Animal Model	Carrier System	Cargo Molecules	Results	Reference
Allergic Asthma	Der f1 induced asthma, mouse	Exosome membrane modified PLGA NP	Antisense oligonucleotides targeting Dnmt3	M2 macrophage silencing, mitigation of lung inflammation	[81]
	OVA induced allergic asthma, mouse	PLGA NP	chrysin	NLRP3 inflammasome repression, reduced IgE and Th2 response	[94]
	OVA induced allergic asthma, mouse	PLGA NP	OVA, A20	Repression of inflammation, increased Treg and IL10	[98]
	OVA induced allergic asthma, mouse	PLGA-PEP-PLGA hydrogel	OVA	Th2, IgE reduction, increased IgG1 titers	[99]
	HDM induced asthma, mouse	PLGA NP, PLGA MP	CpG, Der p2	IgE to IgG2 shift, Th1 response	[101]
Allergic conjunctivitis	Ambrosia artemisiifolia pollen crude protein sensitization, mouse	PEG-PLGA NP	Recombinant Amb a 1	IgE to IgG2 shift, Th1 response	[100]
Cow’s milk allergy	Whey-protein induced CMA, mouse	PLGA NP	β-lactoglobulin derived peptides	Protection from anaphylaxis, preserved immune homeostasis	[103,104]
	β-lactoglobulin sensitization, mouse	PLGA MP	β-lactoglobulin	Decreased IgE titers	[105,106]
Honey bee venom allergy	Bee venom sensitization, mouse	PLGA MP	PLA2, protamine stabilized CpG	Reduced anaphylaxis, IgG2 production, Th1 response	[111]

**Table 3 pharmaceutics-15-00615-t003:** Summary of PLGA mediated immunotherapy of AID.

Disease	Animal Model	Carrier System	Cargo Molecules	Results	Reference
Multiple sclerosis	EAE, mouse	Anti-CD4 mAb coated PLGA NP	LIF	BBB passage, ameliorated disease scores and reduced Il-6 in the frontal cortex	[120]
	EAE, mouse	PLGA NP	MOG_35–55_, IL-10	Ameliorated disease scores, reduced IL-17 and IFNγ production ex vivo	[121]
	EAE, mouse	Dual-sized PLGA MP	MOG_35–55_, TGF-β, GM-CSF	EAE inhibition, ameliorated disease scores, reduced immune cell infiltration	[122]
	EAE, mouse	PLGA NP	MOG_35–55_	Ameliorated disease scores, reduced IL-17 and IFNγ secretion, and increased IL-10 production ex vivo	[123]
	EAE, mouse	PEG-PLGA NP	MOG_35–55_	Ameliorated disease scores, reduced DC activation	[124]
	EAE, mouse	PLGA NP	Glucosamine-MOG_35–55_	Prevention of EAE and disease amelioration	[125]
	EAE, mouse	PLGA NP	PLP_139–151_, TGF-β	Ameliorated disease scores, Treg induction	[127]
	EAE, mouse	PLGA MP	MOG_35–55_, PLP_139–151_	Ameliorated disease scores, reduced IL-17 and IFNγ secretion ex vivo	[128]
	EAE, mouse	PLGA NP	PLP_139–151_	Increased drug and particle dosing, improved disease amelioration and antigen presentation on DCs	[129]
Guillain–Barré Syndrome	Experimental autoimmune neuritis, rat	PLGA NP	Cargo-free	Ameliorated disease scores, reduced expression of IL-17, IFN-γ, TNF-α	[131]
Type I diabetes	NOD, mouse	Dual-sized PLGA MP	B_9–23_, TGF-β, GM-CSF, Vitamin D3	Delayed disease onset, decreased insulitis score, increased Tregs	[133]
	NOD, mouse	PLGA MP, PuraMatrix^TM^ peptide hydrogel	Denatured insulin, CM-GSF, CpG	Delayed onset and disease prevention	[134]
	NOD, mouse	PLGA MP	B_9–23_	Delayed onset and disease prevention	[135]
Rheumatoid arthritis	Collagen-induced arthritis, mouse	PLGA NP	CII	Disease prevention and suppressed disease progression, increased serum TGF-β	[139]
	Collagen-induced arthritis, rat	PLGA MP	Altered CII AP_268–270_	Suppressed disease progression	[140]
	Collagen-induced arthritis, mouse	Membrane microvessicle coated PLGA NP	Tacrolimus	Suppressed disease progression, decreased joint inflammation	[141]
	Collagen-induced arthritis, inflammatory arthritis, mouse	Neutrophil plasma membrane coated PLGA NP	Cargo-free	Disease amelioration, neutralization of TNF-α, IL1-β	[142]
Systemic lupus erythematosus	MAR/lpr, mouse	PEG-PLGA-PLL NP	miRNA-125a	Disease amelioration, increased Treg populations	[146]
	BDF1, mouse	CD2/CD4 mAb coated PLGA-NP	TGF-β, IL-2	Disease amelioration, Treg expansion	[147]

**Table 4 pharmaceutics-15-00615-t004:** Summary of PLGA-mediated immunotherapy of infectious diseases.

Disease	Animal Model	Delivery System	Cargo Molecules	Results	Reference
Viral hepatitis	Mouse	PLGA NP	HBV nucleocapsid core antigen, MPLA	Induction of Th1 response, enhanced response after boost vaccination	[166]
	Rat	PLGA NP	HBsAg	Induction of cellular and humoral response, mucosal IgA	[167]
	Mouse	Mannose modified PLGA NP	HBsAg	Targeted delivery to APCs, induction of cellular and humoral response	[168]
	Mouse	PLGA MP	HCV E2 envelope glycoprotein	Induction of cellular and humoral response	[171]
Influenza	Mouse	PLGA MP	HA (H5N1), imiquimod	Induction of Th1 biased cellular and humoral response	[174]
	Chicken	Chitosan-PLGA NP	Inactivated H4N6, CpG	Induction of humoral response, mucosal IgA	[175]
	Mouse	PLGA NP	Capsomere presenting M2e	Induction of humoral and cellular response	[176]
	AAD transgenic mouse	PLGA MP	M1_58–66_, PA_46–54_, CpG, polyIC	Induction of CTL response, protection from IAV infection	[89]
	Mouse	PLGA MP	OVA, M1_58–66_, CpG, polyIC	Induction of mucosal Trms, protection from IAV infection	[177]
HIV	Guinea pig	PLGA MP	HIV-1 gp120	Induction of neutralizing antibody response	[180]
	Mouse	PLGA MP	PEI/plasmid-DNA complex (Gag, Pol, Env)	Induction of cellular and humoral response	[181]
COVID-19	Mouse	PLGA NP	Recombinant SARS-CoV-2 S1-E	Induction of cellular and humoral response with neutralizing antibodies	[182]
	Mouse	PEG-PLGA implant	Trivalent SARS-CoV-2 subunits	Induction of Th1 biased cellular and humoral response with neutralizing antibodies	[183]

**Table 5 pharmaceutics-15-00615-t005:** Summary of PLGA-mediated cancer immunotherapy.

Therapy	Animal Model	Delivery System	Cargo Molecules	Results	References
Immune checkpoint inhibition	MC38 colon cancer, mouse	Anti-PD-L1 F(ab) conjugated PEG-PLGA NP	Cargo-free	Increased immune cell infiltration into tumor tissue, reduced tumor growth	[191]
PTT + immune checkpoint inhibition	B16/F10 melanoma, mouse	GRGDS-PEG-PLGA NP	Anti-PD1 mAb	Anti-PD-1 mAb release and CD8^+^ T cell infiltration into tumor tissue, prolonged survival	[192]
Modulation of tumor microenvironment	Pan02 pancreatic cancer, mouse	PEG-PLGA NP	Oxaliplatin	DAMP mediated DC and T cell activation in vitro, reduced tumor burden	[195]
Modulation of tumor microenvironment + Immune checkpoint inhibition	4T1 triple negative breast cancer, mouse	Hybrid biomimetic membrane coated PLGA NP	Metformin, siRNA FGL1	Silencing of FGL1 expression, improved tumor-free survival	[196]
Modulation of tumor microenvironment	B16/F10 melanoma, mouse	Tumor cell membrane coated PEP-M2pep-PLGA NP	R848	Reprogramming to M1 TAM, immune cell infiltration into tumor tissue, tumor eradication	[197]
Modulation of tumor microenvironment	B16/F10 melanoma, mouse	PLGA NP, PLGA hydrogel	Paclitaxel, CXCL-1	Infiltration of paclitaxel PLGA NP baring neutrophils into tumor tissue, reduced tumor growth	[201]
Cancer vaccine	B16-OVA melanoma, mouse	PLGA-core/lipid-shell hybrid particles	OVA mRNA, gardiquimod	Induction of CTL response, reduced tumor growth	[207]
Cancer vaccine + immune checkpoint inhibition	B16/F10 melanoma, mouse	DSPE-PEG-mannose modified tumor cell membrane coated PLGA NP	Imiquimod	DC targeting and activation in vitro, tumor growth inhibition and improved tumor free survival, synergy with immune checkpoint inhibition	[208]
Cancer vaccine	4T1 triple negative breast cancer, mouse	Biotin/streptavidin conjugated PLGA MP	4T1 cell lysate	Protection from metastasis formation	[209]
PTT + immune checkpoint inhibition	4T1, CT26 triple negative breast cancer, mouse	PLGA NP, PEG-PLGA NP	Imiquimod, ICG	Primary tumor eradication, inhibition of metastasis formation, synergy with immune checkpoint inhibition	[210]
Cancer vaccine + immune checkpoint inhibition	B16-OVA melanoma, EG7 thymoma, mouse	PLGA NP, PLGA MP	OVA, HLA-A*0201 restricted TAA, Riboxxim	DC activation in vitro, TAA specific CTL responses, protection from metastasis formation, tumor eradication, synergy with immune checkpoint inhibition	[84]
Cancer vaccine + immune checkpoint inhibition	EG7 thymoma, TC-1 myeloma, mouse	PLGA NP	OVA, HPV-E7, siRNA PD1, siRNA PD-L1	Reduction of PD-1, PD-L1 expression, tumor eradication, improved tumor-free survival	[211]
Cancer vaccine	AAD, HHD transgenic mice	PLGA NP	NY-ESO-1 TAA, IMM60	Induction of TAA specific cellular and humoral response, TAA specific activation of human PBMCs in vitro	[212,213]

## Data Availability

Not applicable.

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
