# Peer review of "PLGA Particles in Immunotherapy"

_pharmaceutics, 2023, doi:10.3390/pharmaceutics15020615_

Round 1

Reviewer 1 Report

The review article entitled “Repurposing Drugs: PLGA Particles in Immunotherapy” provided us a relatively comprehensive overview on immunotherapeutic applications of PLGA particles, however, some suggestions for improvements are necessary for future submission:

1.     Repurposing drugs in the title is not reflected in the maintext, so pls. highlight this point in the text or modify the title with more suitable term.

2.     The chemical structure of PLGA should be displayed and its available functional groups for suitable function should also be summarized. As your topic is mainly on PLGA, so its applications in immunotherapy also mainly based on tailored construction of PLGA to form different formulations. The construction strategy should be summarized clearly in tables or other forms.

3.     It is suggested that the large text should be summarized and displayed by matching the appropriate schematic diagram or tables, which will be readable for different readers

4.      “With higher molecular weights, the polymeric chain size increases and so does the degradation rate as a consequence” Pls. check the accuracy of this sentence. The degradation rate becomes slow as the increasing in molecular weight.

5.     Lack of summary in PLGA for different diseases, Tables or Schemes should be used to illustrate the key parameters or characteristics of PLGA that influence the therapy of diseases. 

6.     Lack of summary:Most of the contents belong to the stacking of individual cases,please input your wisdom to summarize with emphasis clearly.

7.     The information of clinical progress on PLGA in different diseases should be involved, which is of great interest to readers.

8.     Some personal evaluation or pros/cons should be involved in some cases listed in the maintext.

Reviewer 2 Report

In this review, the authors introduced and concluded the Physico-chemical properties and application in immunotherapy of PLGA Particles. The logical structure of the manuscript is clear and the article has relatively complete exposition. However, there are several issues need to be revised. Hence, I recommend the manuscript can be published in Pharmaceutics after major revision. Here are my comments and questions.

Major question:

1. The title of this review is “Repurposing Drugs: PLGA Particles in Immunotherapy”. “Repurposing” means “to change somethings slightly in order to make it suitable for a new purpose”. Authors should consider the reasonableness of the application.

2. For a review, a significant character is its clarity. Therefore, an outline in front of the review is very important which can let readers understand the structure and content quickly.

3. As a review, it should be added some summary tables and figures to help the reader understand better.

4. Some surface modification is mentioned in the “Physico-chemical properties of PLGA Particles”, which is obviously not a physico-chemical properties. Authors should re-examine the plausibility of this part.

5. Section three is entitled “Most successful application of PLGA particles”. PLGA is already widely studied. This article does not make a detailed comparison, and authors should consider the validity of the statement.

Other questions:

6. The article can add several typical PLGA preparation methods, which will make the article more complete.

7. At page 1, line 8, “Poly(lactic- co-glycolic)acid (PLGA) particles” is a wrong format.

8. At page 3, line 144, “nervous system (CNS) which, depending on” is a wrong statement and this should be corrected.

9. At page 18, “pictures of PLGA nano- (a)” is a wrong writing in the note of Figure 2 and this should be corrected.

10. “2019. 87, DOI:” in reference 3 is a writing error.

11. There are some mistakes of writing and formatting error in the passage, The authors should check carefully the manuscript again.

Reviewer 3 Report

This manuscript reviewed the application of PLGA particles in immunotherapy. Revisions are suggested based on the following questions.

1. For a review paper, the quality of the figure is relatively low. Also, the number of figures should be increased.

2. The English should be carefully checked. For example, “It is manly treated with anti-inflammatory medication to counteract the immune imbalance [90].” Manly should be mainly?

3. As a component of treatment, the function of PLGA is only focused on DDS. The effect of PLGA itself on the immune responses should be discussed in more detail.

4. There are many other DDS materials, therefore, the merit and demerit of PLGA should be discussed in more detail.

Round 2

Reviewer 1 Report

The revised version is better than before, some minor flaws should be thoroughly checked.

1.     Some abbreviations such as MP,NALP3 etc. appeared at the first time should be specified.

2.     In Fig 1,pls. specify what the triangles and spheres represent?

3.     Some spelling mistakes should be thoroughly checked. Such as “und” in “Though orally administered PLGA particles are subjected to the highly acidic environment of the stomach und subsequently to highly accelerated polymer degradation”

Reviewer 2 Report

No comments or suggestions.

Author Response

Thank you for reviewing our manuscript.

Reviewer 3 Report

The manuscript was carefully revised. 

Author Response

Thank you for reviewing our manuscript.